# Brief communication: The Glacier Loss Day as indicator for a record negative glacier mass balance in 2022

Annelies Voordendag[1], Rainer Prinz[1], Lilian Schuster[1], and Georg Kaser[1]

[1]Department of Atmospheric and Cryospheric Sciences (ACINN), Universität Innsbruck, Austria

**Correspondence:** Annelies Voordendag (annelies.voordendag@uibk.ac.at)

**Abstract.** In the hydrological year 2021/22 Alpine glaciers showed unprecedented mass loss. On Hintereisferner (Ötztal Alps, Austria), the glacier-wide mass balance was -3319 kg m$^{-2}$. Near-daily observations of surface elevation changes from a permanent terrestrial laser scanning setup allowed determining the day when the mass balance of Hintereisferner started to become negative. This Glacier Loss Day (GLD) was already reached on 23 June in 2022 and gave way to a long ice ablation period. In 2021/22, this and the high cumulative positive degree days explain the record mass loss. By comparing the GLDs of 2019/20-2021/22, we found a gross yet expressive indicator of the glacier's imbalance with the persistently warming climate.

## 1 Introduction

During recent decades, climate-induced annual mass balances were persistently negative on most glaciers worldwide (Hugonnet et al., 2021). In the European Alps, the glaciers' mass loss was particularly outstanding in the hydrological year 2021/22 (GLAMOS - Glacier Monitoring Switzerland, 2022; Copernicus Climate Change Service (C3S), 2023) as, in our example, shown in the 70 years long record of annual mass balances on Hintereisferner (HEF, Ötztal Alps, Austria) in Figure 1. The return period for the 2021/22 annual mass balance exceeds 1000 years, even when choosing the most recent 30-year reference period (1992-2021) for the generalised extreme value distribution fit.

Early in the summer 2022 it became obvious that the day would soon be reached when all mass gained from the accumulation period would be lost. By assuming that after this day, during summer, the glacier loses mass irrecoverably for the rest of the mass balance year, we call this day the glacier loss day (GLD) in an approximate analogy to the Earth Overshoot Day (e.g. Wackernagel et al., 2002; Collins et al., 2020), which marks the date when humanity's annual consumption of ecological resources exceeds what Earth can regenerate in the same year. Following a steady state concept, by reaching a GLD, the glacier obviously leaves the balance with the environmental conditions for the given mass balance year. The earlier the GLD is reached, the higher is the probability of massive mass loss. The key to interpret and use the GLD as near real-time gross but expressive indicator for a glacier's state of "illness" is to record its progressing volume and mass loss during an ablation season at daily resolution.

A daily mass loss record of a glacier and the respective detection of a GLD can be retrieved from mass balance modelling in daily time steps at near real-time. Cremona et al. (2023) upscaled the mass loss on three Alpine glaciers from six automated ablation stake readings to the full area of the respective glaciers by applying a mass balance model. The authors do, however,

not explicitly indicate a GLD. A second way is to measure daily volume changes over an entire glacier, as was done on Hintereisferner (HEF) in the three consecutive mass balance years 2019/20, 2020/21 and 2021/22 from daily terrestrial laser scanning (TLS) acquisitions (Voordendag et al., 2021, 2023). From the three years with available data we identify the GLD, and compare the extremely negative mass balance of 2021/22 with the two precedent "normally" negative mass balance years for which near daily TLS data are available. By analyzing these records in the light of ablation season air temperature and winter mass balance records, we explore the GLD's potential for serving as a nearly instant indicator for a glacier's state under the respective year's climatic conditions.

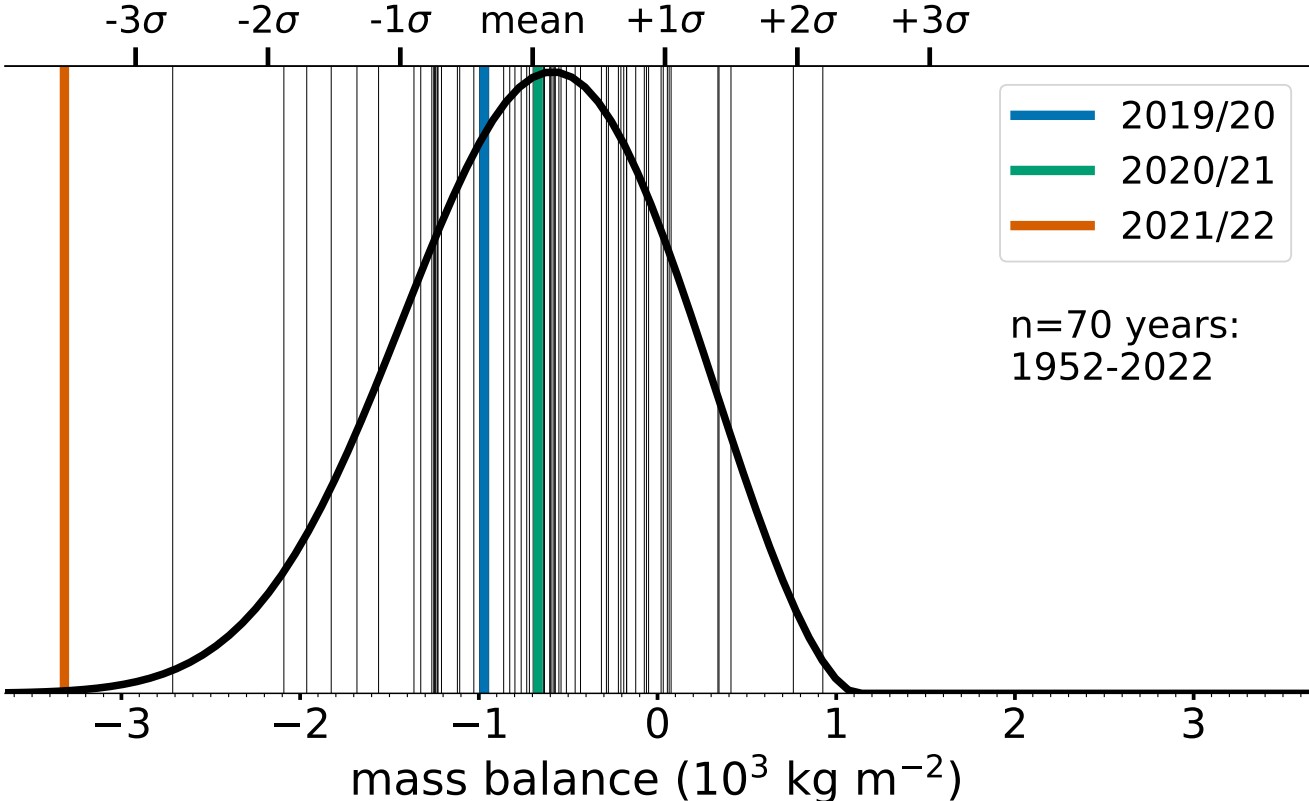

**Figure 1.** The annual mass balance observations of Hintereisferner (in $10^3 \, \mathrm{kg \, m^{-2}}$) over the full 70 years record are depicted as vertical lines. A fitted generalized extreme value distribution of these observations is added to emphasize the extremity of the year 2021/22. The hydrological years used in this study are given in blue (2019/20), green (2020/2021) and red (2021/2022). The idea of this figure was inspired by a Twitter post from Matthias Huss (https://twitter.com/matthias_huss/status/1575539821493293058).

## 2 Data and methods

The well-studied HEF is a valley glacier in the Ötztal Alps, Austria, with continuous mass balance observations since 1952/53
and a rich set of glacier, atmosphere and hydrology data (Strasser et al., 2018), which makes it one of the key 'reference
glaciers' for the World Glacier Monitoring Service (WGMS Zemp et al., 2009).

A permanently installed long-range TLS system *Im Hinteren Eis* (Voordendag et al., 2021) allows the acquisition of a daily
Digital Elevation Model (DEM) of HEF. The TLS system was permanently installed in 2016, has been in daily operational
use since March 2019, and was automated in June 2020 (Voordendag et al., 2021). Additionally, glaciological mass balance
observations with an uncertainty of $\pm 210\,\mathrm{kg\,m^{-2}}$ (Klug et al., 2018) and common meteorological observations from *Station
Hintereis* (3026 m a.s.l.), located on the slope opposite to the TLS system (Strasser et al., 2018), are available since 1952/53.

The data from the TLS is acquired as point clouds and registered manually in the RiSCAN Pro software (RIEGL, 2019). For
this study, it is chosen to calculate grids with a 10 m horizontal grid spacing, where a mean value is taken over all returned laser
pulses in a ten-by-ten metre grid. Albeit grid sizes of one-by-one metre could also be provided from the TLS measurements,
the ten-by-ten metre grid cells are a compromise between high spatial resolution and a good coverage of the glacier. The
spatial coverage increases with a higher horizontal grid spacing. Only few laser pulses are returned per $100\,\mathrm{m^2}$ at long ranges
(>3000 m) and unfavourable incidence angles (Voordendag et al., 2023), so not every one-by-one metre grid cell would be
covered. However, these few pulses are still assumed to be representative for the surrounding area of ten-by-ten metre and with
one-by-one grids, several grid cells would be lost within the $100\,\mathrm{m^2}$. The one-by-one metre DEMs have an accuracy of $\pm 10\,\mathrm{cm}$
(Voordendag et al., 2023) and the same accuracy is assumed for the ten-by-ten metre DEMs. This accuracy is within previously
reported errors of airborne laser scanning acquisitions (Klug et al., 2018) and enables the calculation of volume change rates
and, ultimately, the GLD.

Each DEM is cropped to the 2018 glacier outlines of HEF and the DEM of difference (DoD) between each registered DEM
to the reference DEM of 1 October of the corresponding hydrological year is calculated. Given the scanning geometry of the
system, between 81 and 83% of the glacier area are covered. The unmeasured areas comprise mainly of tributary basins (∼10%)
that are not anymore connected to the glacier and some areas in the upper part of the main glacier. The basins are still included
in the glacier mass balance data of HEF to assure continuity of the mass balance data set (Klug et al., 2018) and are also
included in this study to be able to compare to the glaciological mass balance measurements. 125 usable DoDs are available
in 2019/20, 151 in 2020/21 and 164 in 2021/22 after omitting scans disturbed by clouds or technical problems (e.g., a power
shortage in winter 2019/20 and the impossibility of reaching the setup during the Covid-19 lockdown), and outlier filtering.
No interpolation has been applied to the data set, as interpolation would introduce additional uncertainties. Furthermore, the
coverage seems to have limited influence on the GLD analysis, as our geodetic and glaciological mass balances observations
over the last six hydrological years on HEF agree in their mean seasonal and annual values within the uncertainty range found
by Klug et al. (2018) of $\pm 210$ kg m$^{-2}$. The effects of ice flow divergence (see Cuffey and Paterson, 2010, Sect. 8.5.5) are
included in the surface elevation changes, but these are omitted by integrating the elevation changes over the entire glacier area
(e.g. Kuhn et al., 1999; Zemp et al., 2010; Klug et al., 2018), as we assume with the >91% coverage over the main glacier

and the small influence of the tributary basins. Also, we can neglect the ice dynamics as low flow velocities ($<10$ m year$^{-1}$) suggest limited influence (Stocker-Waldhuber et al., 2019).

Subsequently, the average elevation change weighted over the covered glacier area is calculated and plotted over the hydro-
logical year. The day when the elevation change graphically crosses the reference elevation from the past autumn, is assigned the GLD. As data points may not always be exactly available at the GLD, we interpolate linearly between the last positive and first negative surface elevation change to assign the GLD. Finally, a linear trend is calculated between the first day of the melt season and the GLD, to calculate the melt rate until the GLD is reached. We manually select the first day of the melt season, which is the first day after which surface elevation decrease is continuously observed.

The density for converting daily glacier wide volume into mass change varies over the course of the year between densities for fresh snow and ice. However, as we assume that at the GLD both volume and mass change equal zero, the determination of a conversion density is not necessary. To strengthen this assumption, we compare the accumulation area ratio for years with observed equilibrium mass balances to the AAR at the GLD derived from the TLS. With the extensive glaciological mass balance record at HEF it was found that the accumulation area ratio (AAR) at the end of hydrological years with equilibrium mass balance conditions ($0\pm100$ kg m$^{-2}$) is between 0.63 and 0.75 (mean: 0.69). In the hydrological years of this study, the AAR is observed with the time-lapse cameras and derived from the reflectance images of the TLS (Prantl et al., 2017). Similar AARs at the GLD are found between 0.66 and 0.71 and so, the glacier must be in equilibrium mass balance conditions at the GLD. Moreover, only small inter-annual variations in the snow line retreat pattern occur during the ablation season, indicating a persistence in this correlation.

## 3 Results and discussion

A given GLD is the result of i) the amount of mass gained during the accumulation season starting with 1 October, ii) the start of the ablation season and iii) the strength of the ablation rate until the GLD is reached (Figure 2a).

i) At the end of the accumulation season, a maximum of only 2.1 m of snow had been deposited on average over HEF in 2022 (Figure 2a), which equals a glaciological winter mass balance of only 691 kg m$^{-2}$. In 2020 (2021), the winter accumulation was significantly higher with an average elevation change over HEF of 3.5 m (3.9 m) at maximum, which resulted in a long-term "normal" winter mass balance of 1396 (1219) kg m$^{-2}$. However, the glaciological measurements were corrected to 30 April, whereas the maximum surface elevation changes were found at different dates. The winter mass balance of 2022 was 43% below the decadal average of 2011-2020. Additionally, the day of peak snow height was 29 April in 2022, whereas this was on 7 (28) May for 2020 (2021). Here, we also note that the TLS measurements in 2020 are sparse due to problems with the power supply, and thus the day of peak snow height might have occurred later in May 2020. The low winter mass accumulation in 2022 paved the ground for an early GLD and an extreme glacier mass loss.

ii) In addition, the melting season started earlier in 2022, i.e. on 11 May compared to 23 June 2020 and 28 May 2021, providing a second reason for an extraordinary early GLD. Figure 2b shows the cumulative positive degree days (CPDD)

calculated starting from 1 May with air temperature measurements from the automatic weather station *Station Hintereis*. The first positive degree days (PDD) already occurred on 11 May in 2022, which initiated a nearly uninterrupted melting season. In 2020 and 2021, PDDs were also measured in May, but often followed by cold days halting ablation. The period with continuous significant PDDs started at the end of June in 2020 and at the end of May in 2021.

iii) The surface elevation change rate between the start of the melting season and the GLD for 2020 (2021) was 5.6 (5.2) cm day$^{-1}$, but only 4.4 cm day$^{-1}$ in 2022. The surface elevation change rates until GLD are a combination of compaction of fresh snow and melt. Compaction of fresh snow was likely the case in May/June 2021, as strong snowfall occurred just before this period. The potential for melt in this period is generally a combination of positive air temperatures, global radiation, albedo, clouds, and event-based heat release from rain on snow. In 2022, the lower surface elevation change rate was caused by the early start of the ablation season, when potential melt energy is generally lower due to lower radiation receipts, but high PDDs in that year.

The GLD of the extreme hydrological year 2021/22 is reached on 23 June and on 11 August in 2019/20 and 2020/21. Years with an early GLD allow for more ice exposed to high potential melt energy and a longer ice ablation season and thus, generally offer more negative mass balances. However, the course of the ablation season might be variable and periods with strong melt alternate with periods of no melt or even slight accumulation. Thus, a mass balance prediction from GLD alone seems questionable and mass balance projections must follow scenarios of assumed ablation conditions. In 2022, 100 days remain from the GLD until the end of the hydrological year, whereas only 51 days remain in the two preceding years. The end of the ablation season, which is here defined as the minimum surface elevation change, is almost equal in the three studied years, i.e. on 18 September 2020, on 26 September 2021 and 14 September 2022. Note, that the end of the ablation season is variable, as it can either end with late summer snow fall and/or low temperatures or can extend into the following hydrological year. In 2021/22, the early GLD, enabling a long ice ablation season, and the high CPDD (Figure 2b) explain the record mass loss of 3319 kg m$^{-2}$, which is 3.4 and 5.0 times higher than in the two preceding years (Figure 1) and 3.2 times higher than the long-term mean of 1991-2020.

## 4   Summary and conclusions

Tracking the volume and mass changes of a glacier in daily time steps allows for a nearly instant evaluation of the state of a glacier in a particular year. We introduce the glacier loss day (GLD) as the day in the hydrological year when the mass accumulated during winter is lost and the glacier loses mass irrecoverably for the rest of the mass balance year. Near-daily TLS observations of glacier surface elevation changes at HEF are used and three years of GLD observations are available. The GLD of the extreme hydrological year 2021/22 was already reached on 23 June, whereas this was 49 days later in 2019/20 and 2020/21 on 11 August. In 2021/22, the low winter accumulation and the early start of the ablation season defined the early GLD and gave way to a long and extensive ice ablation period. This and the high CPDD in 2021/22 explained the record mass loss of 3319 kg m$^{-2}$.

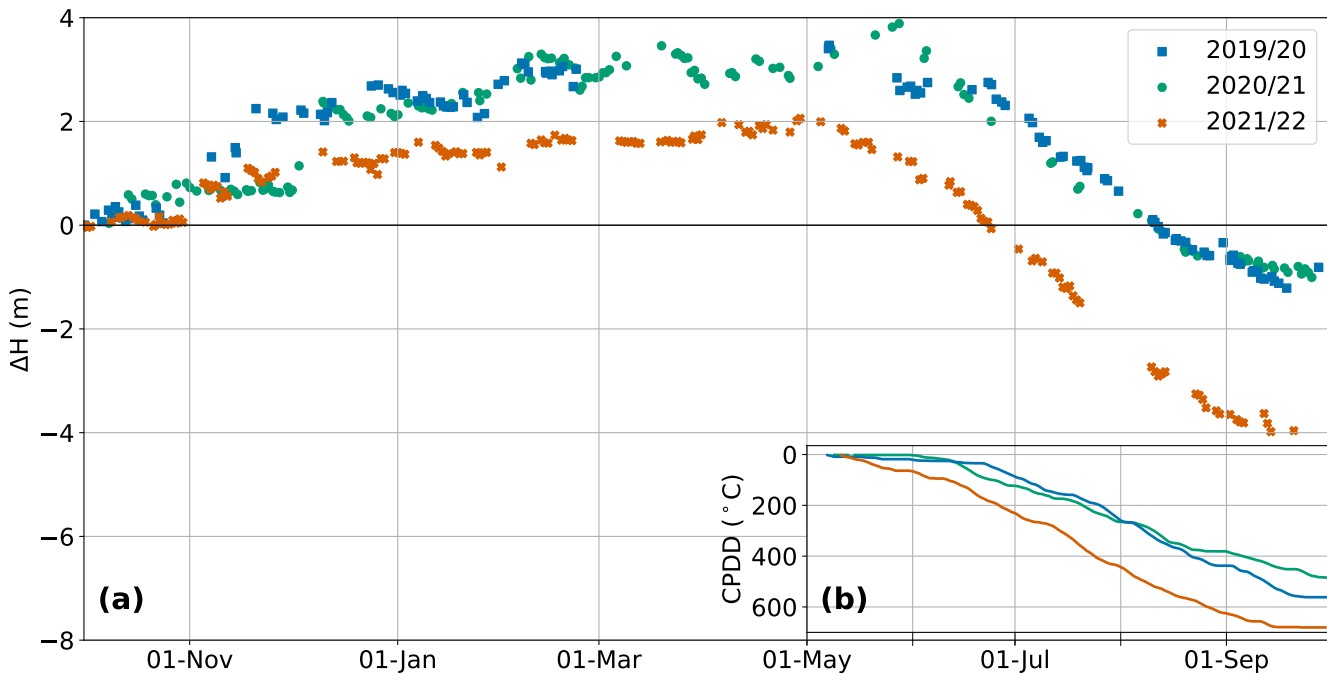

**Figure 2.** (**a**) The average surface elevation change $\Delta H$ over HEF for the hydrological years 2019/20 (blue), 2020/21 (orange) and 2021/22 (green). (**b**) The Cumulative Positive Degree Days (CPDD) calculated from 1 May until 30 September for the same hydrological years.

The GLD summarizes the mass balance development of the respective year and indicates as such the basis for the remaining mass loss potential before accumulation starts. A very early GLD has a high potential for a substantial mass loss year. It contains high public information content and is, similar to the Earth Overshoot Day, a valid and powerful communication tool. Additionally, the GLD could serve for water management strategies and help to schedule scientific field work. However, assumptions must be made on the ablation rates and duration of the ablation season after the GLD has been reached.

Nevertheless, we are aware that a setup such as at HEF is unique and only feasible on specific glaciers. However, with ongoing developments in modelling approaches (e.g. Landmann et al., 2021) and improvements in glaciological and geodetic measurements, mainly with automated ablatometers (e.g. Landmann et al., 2021; Cremona et al., 2023; Carturan et al., 2019; A2 Photonic Sensors, 2022), uncrewed aerial vehicles and the increasing availability of high-resolution satellite data, the GLD of other glaciers can be studied and communicated in the future.

*Code and data availability.* The RiSCAN PRO software is used to register the TLS point clouds and can be obtained from the manufacturer (RIEGL, 2019). The meteorological data of *Station Hintereis* can be retrieved from https://acinn-data.uibk.ac.at/pages/station-hintereis.html. The data and script to reproduce the plots in this article are publicly available via Zenodo: https://doi.org/10.5281/zenodo.7684348.

*Author contributions.* AV registered the TLS point clouds and prepared the data for Figure 2. RP is responsible for the glaciological moni-
toring at HEF. LS provided Figure 1. AV, RP, LS and GK prepared the manuscript together.

*Competing interests.* The authors declare that they have no conflict of interest.

*Acknowledgements.* This research is embedded in the SCHISM project (Snow Cover dynamics and HIgh reSolution Modeling) and is funded
by the Austrian Science Fund (FWF) and the German Research Foundation (DFG) research project I3841-N32 *Snow Cover Dynamics and
Mass Balance on Mountain Glaciers*. LS is recipient of a DOC Fellowship from the Austrian Academy of Sciences (ÖAW). Further funding
was granted by the University of Innsbruck.

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
