# Peer review of "Brief communication: The Glacier Loss Day as indicator for a record negative glacier mass balance in 2022"

_The Cryosphere, 2023_

## Referee Comment (RC2)

The authors present a new indicator for glacier monitoring: the Glacier Loss Day (GLD), i.e. the day of the hydrological year when the mass balance of a glacier becomes negative. They computed the GLD on Hintereisferner for the hydrological years 2020, 2021, and 2022 and discussed differences among the three years with a focus on the extreme mass loss of 2022.

The study is clear, concise and easy to read. In the following, there are a set of minor comments and questions.

**General comments**

L49-54: In this part of the methods, more details on how you derive the elevation change are needed. How do you fill the voids in the DEMs since the average coverages in the DoDs are around 50%? Do you interpolate? If yes, what kind of interpolation?

L59-66: To convert from volume change to mass change is not the "density" necessary, but the "density of volume change". This is something different, since it depends on compaction of the snow/firn layer and its change over time, and not simply, as said, on the surface density of the snow. Please describe better if you use the "density" or the "density of volume change". Furthermore, it is implied (at line 64) that the density distribution is "similar". How do you know that? Are there spatial measurements available, or is it an assumption?

Please provide uncertainty values of the mass balance (e.g. L2 and elsewhere) and in Fig. 2.

**Specific comments**

Introduction

L16-17: "By reaching a GLD, the glacier obviously leaves the state of balance with the environmental conditions." → Is the glacier in balance before the GLD? Or it is actually in balance only at GLD when the mass balance is zero? Or what kind of balance is meant here? Consider rephrasing or omitting.

L21-22: "Cremona et al. (2022) upscaled the mass loss on three Alpine glaciers from automated ablation stake readings at one point to the full glacier area by applying a mass balance model." → Cremona et al. 2022 upscaled mass balance from different point observations, to the scale of the entire Swiss Alps. From this sentence it is not clear 1), weather there is only one or more point observations, and 2) if "the full glacier area" is meant at regional scale or only for the three glaciers. Consider rephrasing.

Fig. 1: It could add value if you make the comparison with other European countries, you could for example reference this link: https://doi.glamos.ch/figures/probability_glacier-wide_annual.pdf
or the GLAMOS data:
GLAMOS (2022). Swiss Glacier Mass Balance, release 2022, Glacier Monitoring Switzerland, doi:10.18750/massbalance.2022.r2022.
https://doi.glamos.ch/data/massbalance/massbalance_2022_r2022.html

Data and Methods

L46: "Given the scanning geometry of the system, around 67% of the glacier area is covered, but this slightly varies over time and glacier conditions." → Which factors cause the area coverage to vary and what is the range in which this vary?

L57-58: "The first day of the melt season is defined as the first maximum after 1 May that is followed by seven consecutive days of surface elevation decrease of at least 4 cm day−1." → Not clear to me the maximum of what. Is it the highest positive mass balance? Consider rephrasing. Is the first day of the melt season defined by you? If so, consider rephrasing to "We defined…". If not, cite literature where this is defined.

L61: "The AAR on HEF at years with equilibrium mass balance conditions of +-100 kg m−2 is 0.69, [...]" → Here you mean the AAR on HEF at GLD right? If so consider rephrasing like "The AAR at the GLD on HEF at years…" or similar.

L62-63: "In the hydrological years of this study, the snow-covered area at the GLD is also approximately 69%, observed from the time-lapse cameras." → How do you compute the snow-covered area? Is it a qualitative estimate or what method do you use?

Results and discussion

L73: "The winter mass balance of 2022 was 47% below the decadal average of 2011-2020." Is the winter mass balance calculated for the same date in both cases? Or how is that homogenized?

Summary and conclusions

L109-110: "In 2021/22, the low winter accumulation, the early start of the ablation season, and the surface elevation change rate define the early GLD and give way to a long ice ablation period." → Actually, the surface elevation change rate before the GLD is lower in 2021/2022 than in previous years. Stated as it is, one could conclude that a lower rate can also contribute to longer ablation period, which is not the case. Consider rephrasing.

L115-119: "Nevertheless, we are aware that a setup such as at HEF is unique and only feasible on specific glaciers. However, with ongoing developments in glaciological and geodetic measurements, mainly with automated ablatometers (e.g. Cremona et al., 2022; Carturan et al., 2019; A2 Photonic Sensors, 2022), uncrewed aerial vehicles and the increasing availability of highresolution satellite data in conjunction with modelling approaches (e.g. Landmann et al., 2021), the GLD of other glaciers can be studied and communicated in the future." → Landmann et al. 2021 developed the stake setup and Cremona et al. 2022 developed the algorithm for the automated reading. Therefore, it is more appropriated to cite Landmann et al. 2021 beforehand (e.g. "…mainly with automated ablatometers (e.g. Landmann et al. 2021; Cremona et al., 2022; Carturan et al., 2019; A2 Photonic Sensors, 2022)"). Furthermore, in Landmann et al. 2021 they don't use UAVs nor satellite data. Please rephrase.

---

## Author Comment (AC1)

**Response to review**
Brief communication: The Glacier Loss Day as indicator for extreme glacier melt in 2022

Annelies Voordendag, Rainer Prinz, Lilian Schuster and Georg Kaser

May 23, 2023

Dear editor and reviewers,

First, we would like to thank the editor and the reviewers for their careful evaluation of our work and the valuable suggestions, comments and questions. We believe that the manuscript has substantially benefited from the reviewers' feedback. Below we address our detailed responses to all the comments.

In this response-to-review document we try to clarify and address each of the suggestions, comments and questions made during the review. Therefore we have copied the comments in blue boxes and have addressed them one by one. In the response we use italic fonts to quote text from the revised manuscript. Additional to the revised manuscript, we have uploaded a version of the manuscript with highlighted track changes that indicate where the manuscript has changed (red=removed; blue=added).

Yours sincerely, Annelies Voordendag & co-authors

**Response to reviewer #1**

**Main comments**

R1-1: The paper employs near-daily observations of surface elevation changes from a permanent terrestrial laser scanning on Hintereisferner (Ötztal Alps, Austria) to retrieve the day when the glacier-wide mass balance becomes negative. Applying the concept during the record-breaking mass loss that occurred in 2022, this Glacier Loss Day (GLD) was reached in late June which gave way to a long ice ablation period. From a comparison with the two previous years, the authors conclude that the GLD can be considered as predictive proxy of the glacier's imbalance for the current hydrological year.

The authors thank the reviewer for its careful evaluation of our work and the valuable suggestions, comments and questions.

R1-2: The paper is concise, almost quite clear and easy to read. The paper would be a bit more complete and understandable were it also to detail how the glacier-wide balance is obtained form observations that only cover a limited surface area (how the spatial interpolation is processed) and how the observed thickness change is converted into a mass change (how vertical velocities are accounted for).

In our study, we do not obtain the glacier-wide mass balance, but we observe surface elevation changes. The surface covered by the TLS does not cover the entire glacier, and initially we calculated grids of one by one meter. That is equal to probing every square meter at a glacier area of $6223923\,\mathrm{m}^2$ (derived from the ALS data acquired by the Federal Government of Tyrol in 2018), compared to the 25 to 40 stakes that are normally used in the glaciological method at HEF. The TLS does not cover this entire area, as some areas, mainly tributaries that are not even connected anymore to the main glacier and some sinks in the accumulation basins at the upper part of the glacier, are excluded due to topographic scan shadows. We do not want to interpolate the data as this also leads to uncertainties and we always cover the same area for every TLS acquisition, which enables us to calculate changes relative to each other. We wanted to utilize the full potential of the high resolution of the TLS system in both space and time, but after the comments of the editor and reviewers, we decided to recalculate the grids at a resolution of 10 by 10 meters.

Often, only one laser pulse per grid cell is returned in the sparsely covered areas. To explain this a bit better, we have made Figure 1 in this review document, where the blue cross is a laser pulse retrieved from the scan at

day $t_1$, the red dot is a laser pulse retrieved from the scan at day $t_2$, the grid size of 1 cell is $1\,\text{m}^2$ and of the entire grid $100\,\text{m}^2$. In the difference plot at a $1\,\text{m}$ grid spacing, the cells with the blue cross and red dot are lost, as they, even though they are close, are not in the same grid cell. However, at a grid spacing of $10\,\text{m}$, we assume that these returned laser pulses are also representative for the surrounding $100\,\text{m}^2$ and thus, these few laser pulses represent the area of $100\,\text{m}^2$ without getting lost when taking the DoD. With this method, more than 51,000 of 62,000 grid cells in the glacier outlines are covered, compared to 25 to 40 stakes in the glaciological method. Thus, the coverage is increased with this method.

Second, our initial calculation method of the mean surface elevation change over all covered cells at $1\,\text{m}$ created a negative bias, as the weight of the calculation is slightly shifted to the glacier tongue, as this area is better covered in comparison to the upper part of the glacier. As a consequence, using the $100\,\text{m}^2$ grids lead to a slightly later GLD. With our new method to grid the point clouds, we cover the main part of HEF, knowing that the missing parts are mainly former tributary basins. These basins are still included in the glacier mass balance data of HEF to assure continuity of the mass balance data set (Klug et al., 2018) and we decided to still include this tributaries, to be able to compare to the glaciological mass balance measurements.

The effects of ice flow divergence (see Cuffey and Paterson, 2010, Sect. 8.5.5) are included in the surface elevation changes. The ice flow divergence is omitted by integrating the elevation changes over the entire glacier area (e.g. Kuhn et al., 1999; Zemp et al., 2010; Klug et al., 2018), as we now assume with the >80% coverage. However, it seems reasonable to neglect the ice dynamics as low flow velocities ($<10$ m year$^{-1}$) suggest limited influence (Stocker-Waldhuber et al., 2019).

[Figure]

Figure 1: Example on how the gridding is done on a $1\,\text{m}$ and $10\,\text{m}$ resolution. For more explanation, please refer to the text.

We have rewritten the method section, and this now reads:
*The well-studied HEF is a valley glacier in the Ötztal Alps, Austria, with continuous mass balance observations since 1952/53 and a rich set of glacier, atmosphere and hydrology data (Strasser et al., 2018), which makes it one of the key 'reference glaciers' for the World Glacier Monitoring Service (WGMS Zemp et al., 2009).*

[revised manuscript text omitted]

The change in the method caused slightly different numbers in the results. We have also adapted the results section. Please refer to the document with differences for this.

> R1-3: A minor point would be to explain what the concept of Glacier Loss Day is primarily (or could potentially be) dedicated to: public communication, current year water resources estimations, glacial hazards... and the degree of approximations and uncertainties accepted for that purpose.

We already stated this in the introduction, where we compare the GLD with the Earth Overshoot Day, and the conclusion ("The GLD contains high public information content and it is, similar to the Earth Overshoot Day, a valid and powerful communication tool. Additionally, the GLD could serve for water management strategies and help to schedule scientific field work."), but we agree that some uncertainties are involved in this. Therefore, we have rewritten this part of the conclusion:
*The GLD summarizes the mass balance development of the respective year and indicates as such the basis for the remaining mass loss potential before accumulation starts. A very early GLD has a high potential for a substantial mass loss year. It contains high public information content and is, similar to the Earth Overshoot Day, a valid and powerful communication tool. Additionally, the GLD could serve for water management strategies and help to schedule scientific field work. However, assumptions must be made on the ablation rates and duration of the ablation season after the GLD has been reached.*

**Detailed notes**

> R1-4: Figure 1. Here you fit annual balances to a Gaussian distribution which is the distribution that fits averages (standard limit theorem). However, in search of extreme mass balance the rareness should be better analysed with extreme values distribution (GEV=Generalized Extreme Value distribution) instead of Gaussian. This should not be too difficult with the very long and highly valuable record of mass balance at HEF.

We have fitted the annual glacier mass balances to the GEV distribution. Fig. 1 is changed and in the caption, it

now reads:

*A fitted generalized extreme value distribution of these observations is added to emphasize the extremity of the year 2021/22.*

R1-5: In connection with the above comment is title of the paper. What was specific in 2022? Extreme melt? Extreme annual balance? Perhaps both. And does extreme mean that 2022 just ranks first? Or does extreme refer to a calculated probability of exceedance?

We decided to change the title "Brief communication: The Glacier Loss Day as indicator for extreme glacier melt in 2022." to:

*Brief communication: The Glacier Loss Day as indicator for a record negative glacier mass balance in 2022.*

We further analysed the "extremity" by calculating the probability of exceedance, as outlined in R1-6.

R1-6: Line 17. Wouldn't an objective of the method be to also estimate for the current year the exceedance probability of the mass loss relative to a given threshold and estimate the rareness of the current mass balance time-plot? I would suggest to use a standard time-plot of mass balance (70-year or 10 last-year standard) reconstructed from the very long series of mass balance at HEF and from a simple degree day model. Here, your analysis is limited by a comparison with the two previous years which is very frustrating knowing the long series of observations at HEF.

The scope of this brief communication study was not to model the annual or seasonal mass-balance with a degree-day model, as mentioned in your comment. We decided only to label the years 2020, 2021 and 2022, because these are the years where the TLS data are available.

However, we have now estimated the rareness of the extreme negative 2022 event by computing the return period for that event under different 30-year reference time periods of annual mass-balance between 1953 and 2021. For each reference period, we fitted a generalized extreme value distribution. The resulting return period for the 2022 annual MB value was well above 10000 years for the oldest 30-year reference periods, i.e., from 1953-1982 until 1977-2006. Even for the most recent 30-year reference period (1991-2022), the return period of that extremely negative annual MB was above 1000 years (corresponding to an exceedance probability rate of $<0.1\%$). However, we needed to assume stationarity to fit the generalised extreme value distribution over the annual MB time series, although we know that even in the 30-year period, there is a trend towards a more negative annual mass balance over time. To get a rough estimate, this approach is still valid. Thus, we added the following to the manuscript:

*The return period for the 2021/22 annual mass balance exceeds 1000 years, even when choosing the most recent 30-year reference period (1992-2021) for the generalised extreme value distribution fit.*

R1-7: Line 25. Not clear what "analogous" means. Do you refer here to the density conversion? This assumption is not just a density conversion but strongly depends whether the volume change is measured on the overall glacier surface or not.

Do you set that assimilating a local thickness change to a wide mass balance is sufficient for the GLD approach? This should be established using the three years of observation (or on synthetic data) with an error assessment to conclude.

In this study, we don't apply a mass conversion to volume changes, as subdaily information on the density is unavailable, and we also stated that the assumption that mass is equal to the volume is only valid at the GLD. We changed the text to:

*From the three years with available data we identify the GLD, and compare the extremely negative mass balance of 2021/22 with the two precedent "normally" negative mass balance years for which near daily TLS data are available.*

In the methods section, we explain why the assumption that a mass balance of $0\,\mathrm{kg\,m^{-2}}$ equals $0\,\mathrm{m}$ surface elevation change at the GLD. This has also been adjusted:

*The density for converting daily glacier wide volume into mass change varies over the course of the year between densities for fresh snow and ice. However, as we assume that at the GLD both volume and mass change equal zero, the determination of a conversion density is not necessary. To strengthen this assumption, we compare the accumulation area ratio for years with observed equilibrium mass balances to the AAR at the GLD derived from the TLS. With the extensive glaciological mass balance record at HEF, it was found that the accumulation area ratio (AAR) at the end of hydrological years with equilibrium mass balance conditions ($0\pm100$ kg m$^{-2}$) is between 0.63 and 0.75 (mean: 0.69). In the hydrological years of this study, the AAR is observed with the time-lapse cameras and*

*derived from the reflectance images of the TLS (Prantl et al., 2017). Similar AARs at the GLD are found between 0.66 and 0.71 and so, the glacier must be in equilibrium mass balance conditions at the GLD. Moreover, only small inter-annual variations in the snow line retreat pattern occur during the ablation season, indicating a persistence in this correlation.*

> R1-8: Lines 20-25. Here you point two majors issues:
>
> - How get a glacier-wide balance balance from a point measurement, or at least a mass balance estimated on a limited surface area
>
> - How convert a thickness change into a mass change accounting for vertical velocities.
>
> In the section 2 (Data&Method), you may explain in more details how you solve these 2 points.

We refer to this in comment R1-2.

> R1-9: Line 50. There is a large surface which is not observed when you cross the different surveys. How to you get a glacier-wide mass change? Do you use a flow model to account for the glacier dynamics? As far as I understand the process, you need to first estimate the surface mass balance in the area where you scan with a flow model to correct from vertical velocity, and then extrapolate the mass balance to the overall glacier surface from the spatial dependence of the mass balance (altitudinal gradient or something more complex).

We use the surface elevation changes as proxy for the glacier mass changes and state that this is only valid at the GLD. Also, the measurements with the TLS are not a point measurement. In comparison to the glaciological method, where 10 to 50 stakes are used, the 43 million points measured with the TLS at HEF and its surroundings (Voordendag et al., 2021) is already a big improvement. Please also refer to R1-2.

> R1-10: Figure 2. It is not clear to me if it is the glacier-wide thickness change in which case it would more be convenient to plot the glacier-wide mass balance applying a density conversion. I would suggest to plot in different slots the local thickness change, the local mass change, and then the glacier-wide change compared to a standard plot (70-year or 10 last-year standard).

We cannot convert the glacier-wide surface elevation changes to mass changes, as density measurements are only available at the dates when the glaciological mass balance is measured, i.e. twice a year. A density conversion would introduce other uncertainties and therefore, we would like to remain with the surface elevation changes. We only show the average surface elevation changes and not local changes as we want to keep the message simple and within the scope of a brief communication.

**Response to the review of Aaron Cremona**

**Main comments**

> R2-0: The authors present a new indicator for glacier monitoring: the Glacier Loss Day (GLD), i.e. the day of the hydrological year when the mass balance of a glacier becomes negative. They computed the GLD on Hintereisferner for the hydrological years 2020, 2021, and 2022 and discussed differences among the three years with a focus on the extreme mass loss of 2022. The study is clear, concise and easy to read. In the following, there are a set of minor comments and questions.

The authors thank Aaron Cremona for his evaluation of our work and the valuable suggestions, comments and questions.

> R2-1: L49-54: In this part of the methods, more details on how you derive the elevation change are needed. How do you fill the voids in the DEMs since the average coverages in the DoDs are around 50%? Do you interpolate? If yes, what kind of interpolation?

We referred to this comment in R1-2.

R2-2: L59-66: To convert from volume change to mass change is not the "density" necessary, but the "density of volume change". This is something different, since it depends on compaction of the snow/firn layer and its change over time, and not simply, as said, on the surface density of the snow. Please describe better if you use the "density" or the "density of volume change". Furthermore, it is implied (at line 64) that the density distribution is "similar". How do you know that? Are there spatial measurements available, or is it an assumption?

The editor made a similar comment on this: "A strong assumption is that the time of 0 volume loss is also the time of 0 mass loss. This is discussed in the paper but I did not fully understand the arguments. This can certainly be clarified."

We have rewritten this section and emphasize more strongly that the relation of $0\,\text{m}$ surface elevation change equals $0\,\text{kg}\,\text{m}^{-2}$ only holds at the GLD:

*The density for converting daily glacier wide volume into mass change varies over the course of the year between densities for fresh snow and ice. However, as we assume that at the GLD both volume and mass change equal zero, the determination of a conversion density is not necessary. To strengthen this assumption, we compare the accumulation area ratio for years with observed equilibrium mass balances to the AAR at the GLD derived from the TLS. With the extensive glaciological mass balance record at HEF it was found that the accumulation area ratio (AAR) at the end of hydrological years with equilibrium mass balance conditions ($0\pm100$ kg $m^{-2}$) is between 0.63 and 0.75 (mean: 0.69). In the hydrological years of this study, the AAR is observed with the time-lapse cameras and derived from the reflectance images of the TLS (Prantl et al., 2017). Similar AARs at the GLD are found between 0.66 and 0.71 and so, the glacier must be in equilibrium mass balance conditions at the GLD. Moreover, only small inter-annual variations in the snow line retreat pattern occur during the ablation season, indicating a persistence in this correlation.*

R2-3: Please provide uncertainty values of the mass balance (e.g. L2 and elsewhere) and in Fig. 2.

Fig. 2 shows the surface elevation changes, which have an uncertainty of $\pm10\,\text{cm}$ (Voordendag et al., 2023). This had already been stated in the section "Data and methods". We have added the glaciological mass balance observations with uncertainties to the section "Data and methods" as well:

*Additionally, glaciological mass balance observations with an uncertainty of $\pm210\,\text{kg}\,\text{m}^{-2}$ (Klug et al., 2018) since 1952/53 and common meteorological observations from Station Hintereis (3026 m a.s.l.), located on the slope opposite to the TLS system (Strasser et al., 2018), are available.*

**Specific Comments**

R2-4: L16-17: "By reaching a GLD, the glacier obviously leaves the state of balance with the environmental conditions." → Is the glacier in balance before the GLD? Or it is actually in balance only at GLD when the mass balance is zero? Or what kind of balance is meant here? Consider rephrasing or omitting.

Following a steady state concept, the glacier is in balance with the environmental conditions at zero mass or volume change. A minimum time frame for this concept is an annual cycle to allow a common reference between glacier and environment as a stratigraphic layer or fixed date, here 1 October. Hence, there is no GLD in positive mass balance years. The GLD for an equilibrium mass balance year must be 30 September (or a bit earlier in the case of some September accumulation) or it occurs much earlier in negative mass balance years. It is irrelevant what the (transient) mass balance might be before the GLD, we just argue that, if GLD is reached well before the reference date, the glacier is inevitably out of balance, because in a winter type accumulation regime net accumulation in the ablation season is not expected. We clarify by rephrasing:

*Following a steady state concept, by reaching a GLD, the glacier obviously leaves the balance with the environmental conditions for the given mass balance year.*

R2-5: L21-22: "Cremona et al. (2023) upscaled the mass loss on three Alpine glaciers from automated ablation stake readings at one point to the full glacier area by applying a mass balance model." → Cremona et al. (2023) upscaled mass balance from different point observations, to the scale of the entire Swiss Alps. From this sentence it is not clear 1), weather there is only one or more point observations, and 2) if "the full glacier area" is meant at regional scale or only for the three glaciers. Consider rephrasing.

First, we note that the preprint of Cremona et al. (2022) is now published and we adjusted that throughout the

text. We rephrased the mentioned sentence to:

*Cremona et al. (2023) upscaled the mass loss on three Alpine glaciers from six automated ablation stake readings to the full area of the respective glaciers by applying a mass balance model.*

> **R2-6:** Fig. 1: It could add value if you make the comparison with other European countries, you could for example reference this link: `https://doi.glamos.ch/figures/probability_glacier-wide_annual.pdf` or the GLAMOS data:

We have removed the footnotes to the newspaper articles and replaced it by the suggested reference and the report of the European state of the climate 2022 by the Copernicus Climate Change Services:

*In the European Alps, the glaciers' mass loss was particularly outstanding in the hydrological year 2021/22 (GLAMOS - Glacier Monitoring Switzerland, 2022; Copernicus Climate Change Service (C3S), 2023) (..)*

> **R2-7:** L46: "Given the scanning geometry of the system, around 67% of the glacier area is covered, but this slightly varies over time and glacier conditions." → Which factors cause the area coverage to vary and what is the range in which this vary?

We refer to this in R1-2. The coverage was mainly reduced because of topographic scan shadows. Thus, it is always approximately the same area that is not covered and we have improved the coverage by using grid cells of 10 by 10 m.

> **R2-8:** L57-58: "The first day of the melt season is defined as the first maximum after 1 May that is followed by seven consecutive days of surface elevation decrease of at least $4\,\mathrm{cm\ day^{-1}}$." → Not clear to me the maximum of what. Is it the highest positive mass balance? Consider rephrasing. Is the first day of the melt season defined by you? If so, consider rephrasing to "We defined...". If not, cite literature where this is defined.

Our method to select the first day of the melt season was not very robust. Therefore we decided to select the day manually and adjusted this in the text:

*We manually select the first day of the melt season, which is the first day after which surface elevation decrease is continuously observed.*

> **R2-9:** L61: "The AAR on HEF at years with equilibrium mass balance conditions of $\pm100\,\mathrm{kg\ m^{-2}}$ is 0.69,[...]" → Here you mean the AAR on HEF at GLD right? If so consider rephrasing like "The AAR at the GLD on HEF at years..." or similar.

We have rewritten this paragraph. Please refer to R2-2.

> **R2-10:** L62-63: "In the hydrological years of this study, the snow-covered area at the GLD is also approximately 69%, observed from the time-lapse cameras." → How do you compute the snow-covered area? Is it a qualitative estimate or what method do you use?

As stated by Prantl et al. (2017), we can distinguish between snow and ice with the reflectance images of the TLS. We have also done this for the studied years with the TLS acquisition closest to GLD. To do so, we calculated all the pixels that have the reflectance values of ice and divided this amount of pixels by the total area of the glacier outlines. With that, we assume that the areas that were not covered by the TLS are still snow-covered, which we know from field trips. Furthermore, the webcam images confirmed our assumption. Please also refer to R2-2.

> **R2-11:** L73: "The winter mass balance of 2022 was 47% below the decadal average of 2011-2020." Is the winter mass balance calculated for the same date in both cases? Or how is that homogenized?

First, we made a typo. 47% should be 43%. This has been adjusted in the text. Second, the mass balance monitoring on Hintereisferner follows a fixed date system. Winter mass balance measurements are carried out around and corrected to 30 April. Corrections make use of nearby meteorological measurements and a degree day approach, but never exceeded $\pm80\,\mathrm{kg\ m^{-2}}$. Stratigraphically, maximum snow depths may differ from this date, e.g. in 2021, Fig. 2, highlighting the importance of daily observations to explain mass loss as cumulative signal of winter accumulation and summer ablation. We added "glaciological" to winter mass balance in the sentence, to show that we only refer to glaciological mass balances and also added that the glaciological measurements are corrected to 30 April:

*At the end of the accumulation season, a maximum of only 2.1 m of snow had been deposited on average over HEF in 2022 (Figure 2a), which equals a glaciological winter mass balance of only 691 kg m$^{-2}$. In 2020 (2021), the winter accumulation was significantly higher with an average elevation change over HEF of 3.5 m (3.9 m) at maximum, which resulted in a long-term "normal" winter mass balance of 1396 (1219) kg m$^{-2}$. However, the glaciological measurements were corrected to 30 April, whereas the maximum surface elevation changes were found at different dates.*

> R2-12: L109-110: "In 2021/22, the low winter accumulation, the early start of the ablation season, and the surface elevation change rate define the early GLD and give way to a long ice ablation period." → Actually, the surface elevation change rate before the GLD is lower in 2021/2022 than in previous years. Stated as it is, one could conclude that a lower rate can also contribute to longer ablation period, which is not the case. Consider rephrasing.

We removed the surface elevation change rate from this sentence and the text now reads:
*In 2021/22, the low winter accumulation and the early start of the ablation season defined the early GLD and gave way to a long and extensive ice ablation period.*

> R2-13: L115-119: "Nevertheless, we are aware that a setup such as at HEF is unique and only feasible on specific glaciers. However, with ongoing developments in glaciological and geodetic measurements, mainly with automated ablatometers (e.g. Cremona et al., 2023; Carturan et al., 2019; A2 Photonic Sensors, 2022), uncrewed aerial vehicles and the increasing availability of high-resolution satellite data in conjunction with modelling approaches (e.g. Landmann et al., 2021), the GLD of other glaciers can be studied and communicated in the future." → Landmann et al. (2021) developed the stake setup and Cremona et al. (2023) developed the algorithm for the automated reading. Therefore, it is more appropriated to cite Landmann et al. (2021) beforehand (e.g. "...mainly with automated ablatometers ((e.g. Landmann et al., 2021; Cremona et al., 2023; Carturan et al., 2019; A2 Photonic Sensors, 2022)"). Furthermore, in Landmann et al. (2021) they don't use UAVs nor satellite data. Please rephrase.

We rephrased the text:
*However, with ongoing developments in modelling approaches (e.g. Landmann et al., 2021) and improvements in glaciological and geodetic measurements, mainly with automated ablatometers (e.g. Landmann et al., 2021; Cremona et al., 2023; Carturan et al., 2019; A2 Photonic Sensors, 2022), uncrewed aerial vehicles and the increasing availability of high-resolution satellite data, the GLD of other glaciers can be studied and communicated in the future.*

---

## Author Response (AR2)

**Response to review**
Brief communication: The Glacier Loss Day as indicator for extreme glacier melt in 2022

Annelies Voordendag, Rainer Prinz, Lilian Schuster and Georg Kaser

July 20, 2023

Dear editor and reviewers,

First, we would like to thank the editor and the reviewers for their second evaluation of our work. We have now implemented the minor requested changes.

In this response-to-review document we try to clarify and address each of the suggestions, comments and questions made during the review. Therefore we have copied the comments in blue boxes and have addressed them one by one. In the response we use italic fonts to quote text from the revised manuscript. Additional to the revised manuscript, we have uploaded a version of the manuscript with highlighted track changes that indicate where the manuscript has changed (red=removed; blue=added).

Yours sincerely, Annelies Voordendag & co-authors

**Response to reviewer #1**

**Minor comments**

> R1-0: The second version of the paper is now much more clear and improved, specifically regarding potential confusions for the reader between thickness changes and mass balance . In addition, the limitations form the limited observed area with the TLS are better discussed. I have only minor and technical points to mention.

The authors thank the reviewer for its second evaluation of our work.

> R1-1: Fig. 1 and associated text. Your fit seems to result in a Gumbel (no lower bounded) distribution. Please specify, and clarify (in the figure caption) if "mean" and sigma are the position of the centre of the distribution and the scale parameters, respectively or still denote the Gaussian mean and SD.

We have fitted a generalised extreme value (GEV) distribution, not a left-skewed Gumbel distribution. That means that the shape parameter is not zero in the fit and that the distribution is unbounded on the lower and upper end (fitted parameters of the GEV: shape=0.380, location=-0.926, scale=0.775). As far as we understand, the GEV is more suited here as the mass balance estimates could be extreme in both directions (very negative and very positive), although we can see that the GEV is rather left-skewed. On the secondary y-axis, we now show the GEV's median and other quantiles instead of the GEV's mean and sigma. This is more meaningful as the GEV is skewed to the left. In addition, with the quantiles of the GEV on the secondary y-axis, it is now clearer that the 2021/22 annual MB value is more negative than the 0.1%-quantile of the GEV, which also shows visually that the return period exceeds 1000 years. We clarified this in the caption of the figure:
*A fitted generalized extreme value distribution of these observations is added with the respective quantiles on the secondary x-axis to emphasize the extremity of the year 2021/22.*

> R1-2: Line 12: You should better consider the complete mass balance record (and not the last 30-year record) as a 1000-year return period is a high estimation (14 times the HEF mass balance period of record).

With the 70-year mass balance series, we also found a return period of more than 1000 years. We have rewritten the text:
*The return period for the 2021/22 annual mass balance exceeds 1000 years when considering the 70-year mass balance series for the generalised extreme value distribution fit.*

> R1-3: Lines 67-68. Do you refer to mass fluxes at the boundaries of your integration area? Reformulate if necessary.

We reformulated this sentence:

*The effects of ice flow divergence on the surface elevation changes (see Cuffey and Paterson, 2010, Sect. 8.5.5) are omitted as ice flow velocities are very low ($<10$ m year$^{-1}$ (Stocker-Waldhuber et al., 2019) and by integrating the elevation changes over the entire glacier area (e.g. Kuhn et al., 1999; Zemp et al., 2010; Klug et al., 2018).*

**Script and spelling**

> Title and in the text, maybe use: « record-breaking » instead of « record »?

We decided to remain with "record".

> Degree day or degree-day? Snow fall or snowfall?

In both cases, both are possible. In our text we consequently used "degree day" and "snowfall".

**Response to the review of Aaron Cremona**

> R2-1: The authors implemented successfully all but one comment, i.e. R2-5. "Cremona et al. (2023) upscaled the mass loss on three Alpine glaciers from six automated ablation stake readings to the full area of the respective glaciers by applying a mass balance model." As already specified in the first round of review, Cremona et al. (2023) upscaled mass balance from six point observations not only to the full area of the respective glaciers but to the scale of the entire Swiss Alps.

We have adjusted this and the text now reads:

*Cremona et al. (2023) upscaled the mass loss on three Alpine glaciers from six automated ablation stake readings to the full area of the respective glaciers by applying a mass balance model, and eventually also upscaled this to the scale of the entire Swiss Alps. The authors do, however, not explicitly indicate a GLD for these glaciers.*

> R2-2: From my perspective, the paper can be accepted once this minor revision is corrected.

We thank Aaron Cremona for his evaluation of our revised manuscript and hope that the manuscript is now fit for publication.

**References**

Cremona, A., Huss, M., Landmann, J. M., Borner, J., and Farinotti, D.: European heat waves 2022: contribution to extreme glacier melt in Switzerland inferred from automated ablation readings, The Cryosphere, 17, 1895–1912, doi: 10.5194/tc-17-1895-2023, 2023.

Cuffey, K. M. and Paterson, W. S. B.: The Physics of Glaciers, Elsevier LTD, Oxford, 2010.

Klug, C., Bollmann, E., Galos, S. P., Nicholson, L., Prinz, R., Rieg, L., Sailer, R., Stötter, J., and Kaser, G.: Geodetic reanalysis of annual glaciological mass balances (2001–2011) of Hintereisferner, Austria, The Cryosphere, 12, 833–849, doi: 10.5194/tc-12-833-2018, 2018.

Kuhn, M., Dreiseitl, E., Hofinger, S., Markl, G., Span, N., and Kaser, G.: Measurements and Models of the Mass Balance of Hintereisferner, Geografiska Annaler, Series A: Physical Geography, 81, 659–670, doi: 10.1111/j.0435-3676.1999.00094.x, 1999.

Stocker-Waldhuber, M., Fischer, A., Helfricht, K., and Kuhn, M.: Long-term records of glacier surface velocities in the Ötztal Alps (Austria), Earth System Science Data, 11, 705–715, doi: 10.5194/essd-11-705-2019, 2019.

Zemp, M., Jansson, P., Holmlund, P., Gärtner-Roer, I., Koblet, T., Thee, P., and Haeberli, W.: Reanalysis of multi-temporal aerial images of Storglaciären, Sweden (1959–99) – Part 2: Comparison of glaciological and volumetric mass balances, The Cryosphere, 4, 345–357, doi: 10.5194/tc-4-345-2010, 2010.

---

## Author Response (AR3)

**Response to review**
**Brief communication: The Glacier Loss Day as indicator for extreme glacier melt in 2022**

Annelies Voordendag, Rainer Prinz, Lilian Schuster and Georg Kaser

August 1, 2023

Dear editor,

Thank you very much for accepting our article for publication. We have adjusted your final three remarks.

In this response-to-review document we try to clarify and address each of the suggestions, comments and questions made during the review. Therefore we have copied the comments in blue boxes and have addressed them one by one. In the response we use italic fonts to quote text from the revised manuscript. Additional to the revised manuscript, we have uploaded a version of the manuscript with highlighted track changes that indicate where the manuscript has changed (red=removed; blue=added).

Yours sincerely, Annelies Voordendag & co-authors

**Response to the editor**

**Minor comments**

> E1: L67. "the average elevation change weighted over the covered glacier area is calculated". The statement was not clear to me, what weight and how they are applied. I think it is important to be clear on how the elevation change measurements, obtained on a 100 $m^2$ grid covering 80% of the glacier area, are averaged to compute a glacier-wide value. In order to clarify for the readers how the roughly 20% missing area are treated.

We explained in L54-58 that the uncovered part of the glacier is mainly attributed to tributary basins, but we understand that "weighted over the covered area" caused some confusion. We meant to say, that as the covered area per scan is slightly different each time, the weight is attributed to the amount of covered grid cells. We have rewritten the sentence to:

*Subsequently, the average elevation change of the covered glacier area is calculated and plotted over the hydrological year.*

> E2:L86. Here you refer to a maximum in the time series. When I read first I thought you provided the maximum in the spatial pattern. My suggestion to avoid this ambiguity: "At the end of the accumulation season, the average (or glacier-wide) elevation change peak at a maximum of only 2.1 m of snow"

We adjusted this according to your suggestion:

*At the end of the accumulation season, the average (or glacier-wide) elevation change peaks at a maximum of only 2.1 m of snow in 2022 (Figure 2a),*

> E3: L137. You may want to add a reference for the availability of very high resolution satellite data as you have references for the others statements.

We added a reference to Pandey et al. (2022):

*(..) and the increasing availability of high-resolution satellite data (Pandey et al., 2022), (..)*

**References**

Pandey, M., Pandey, P. C., Ray, Y., Arora, A., Jawak, S. D., and Shukla, U. K., eds.: Advances in Remote Sensing Technology and the Three Poles, Wiley, doi: 10.1002/9781119787754, 2022.